# The Effects of Artificial Diets on the Expression of Molecular Marker Genes Related to Honey Bee Health

**DOI:** 10.3390/ijms25084271

**Published:** 2024-04-12

**Authors:** Olga Frunze, Hyunjee Kim, Jeong-Hyeon Lee, Hyung-Wook Kwon

**Affiliations:** 1Department of Life Sciences, Incheon National University, 119 Academy-ro, Yeonsu-gu, Incheon 22012, Republic of Korea; frunzeon@gmail.com (O.F.); jeonghyeon@inu.ac.kr (J.-H.L.); 2Convergence Research Center for Insect Vectors (CRCIV), Incheon National University, 119 Academy-ro, Yeonsu-gu, Incheon 22012, Republic of Korea

**Keywords:** *Apis mellifera*, insect model, nutrition, water-soluble proteins, innate immunity, winter honey bees, early spring, bee bread, functional food

## Abstract

Honey bees are commonly used to study metabolic processes, yet the molecular mechanisms underlying nutrient transformation, particularly proteins and their effects on development, health, and diseases, still evoke varying opinions among researchers. To address this gap, we investigated the digestibility and transformation of water-soluble proteins from four artificial diets in long-lived honey bee populations (*Apis mellifera ligustica*), alongside their impact on metabolism and DWV relative expression ratio, using transcriptomic and protein quantification methods. Diet 2, characterized by its high protein content and digestibility, was selected for further analysis from the other studied diets. Subsequently, machine learning was employed to identify six diet-related molecular markers: *SOD1*, *Trxr1*, *defensin2*, *JHAMT*, *TOR1*, and *vg.* The expression levels of these markers were found to resemble those of honey bees who were fed with Diet 2 and bee bread, renowned as the best natural food. Notably, honey bees exhibiting chalkbrood symptoms (Control-N) responded differently to the diet, underscoring the unique nutritional effects on health-deficient bees. Additionally, we proposed a molecular model to elucidate the transition of long-lived honey bees from diapause to development, induced by nutrition. These findings carry implications for nutritional research and beekeeping, underscoring the vital role of honey bees in agriculture.

## 1. Introduction

Honey bees produce many products for human health and play a crucial ecological role in both natural ecosystems and agriculture. Similar to other insects, honey bees present both an alternative source of protein for humans and a convenient animal model for fields from genetics to neurobiology, especially for the early stage preclinical research of pathogen infection experiments, innate immune system research, and nutrition effects [1,2].

These benefits of honey bees are available because insects and humans possess some equivalent organs and similar biological systems, including the evolutionarily conserved gut and digestive system [1]. Malnutrition, overfeeding, high-sugar diet (HSD), and high-fat diet (HFD) effects on metabolism in a *Drosophila* model were described [3,4,5], with a focus on their effects on lifespan, health, age-related diseases, and aging [6,7,8]. Next, nutrition, proteins in the diet, and their transformation from food to organism are critical, as animals construct their bodies from amino acids derived from animal and plant foods, which are subsequently passed through the food chain. For instance, proteins are responsible for the development of the fat body in the abdomen of individual honey bees, influencing energy storage, production, and the secretion of vitellogenin (vg) protein, antimicrobial peptides, and components of the humoral immune response [9]. Despite the obvious importance of proteins for animals, our understanding of their effects on health and development still contains gaps. To study this, a model sensitive to protein deficiency needs to be chosen from among the many available insect models.

In this context, the health of a honey bee colony primarily relies on proteins collected by forager honey bees from plant sources, such as flower pollen, and their subsequent storage in the hive in the form of bee bread [10]. This food, together with nectar, supports the development and maintenance of a normal immune system [11], as they supplement honey bees not only with proteins and carbohydrates, but also hydrophilic antioxidants and other necessary nutrients [12]. In honey bees, colonies without pollen supply maintain brood rearing only for a short time, first by using up the stored bee bread and later by depleting their body reserves [13]. If the pollen dearth continues, non-foraging honey bees engage in the cannibalism of larvae younger than three days old [14], and no more brood can be produced [6], because pollen ingestion is necessary to develop hypopharyngeal glands in the head [15,16]. So, honey bees seem to be a good model to study the protein-rich needs and deficiencies of animals.

Moreover, beekeepers and agriculturists need to develop artificial (pollen substitute) diets for honey bees to overcome the scarcity of pollen sources in nature during early spring, dry summers, and cold fall months [17,18,19]. This scarcity has led to honey bee starvation and increasing colony losses worldwide [8]. To overcome pollen dearth times, beekeepers use a multitude of different pollens, artificial diets, and feeding practices, but there is no robust research that universally supports their benefits for the health of honey bees [20]. Moreover, the given pollen is expensive and can spread disease [21,22]. This situation emphasizes the significance of developing pollen substitute diets for honey bees, especially given the established relationship between nutrition and the immune system of animals [23,24,25].

Nutrients play a crucial role in controlling the expression of cytokines via the Toll pathway, thereby affecting neutralized cytotoxicity [26]. Conversely, a decline in protein metabolism, associated with the diminishing concentration of certain amino acids, can lead to endoplasmic reticulum (ER) stress and the activation of pro-inflammatory cytokines [27]. Notably, increasing dietary protein intake improved the immune status of one of two pig breeds, as evidenced by changes in immunoglobulin and cytokine levels [23]. In experiments with caterpillars of the insect *Spodoptera littoralis*, non-immune genes (*Arylphorin*, *EF1*, *Armadillo*, and *Tubulin*) showed a consistently weak response to dietary manipulation. In contrast, immune genes such as *Toll*, *Prophenoloxidase*, and *Lysozyme* (Toll pathway) and *Moricin* and *Relish* (Imd pathway) genes exhibited stronger responses [24]. Despite the considerable interest in the interaction between dietary protein and immunity, there are still areas of limited knowledge regarding standard molecular effects and nutrient markers to identify the best diet.

To address these gaps, we conducted a field experiment on overwintered *Apis mellifera ligustica* honey bees in early spring, when natural sources of nectar and pollen are scarce. Therefore, this study aims to examine the impacts of different types of pollen substitution diets (Diet 1, Diet 2, and Control-N), the commercial diet supplement Megabee, and bee bread (Control-P) based on several aspects: (1) the relationship between water-soluble protein content in diets and protein digestion in honey bees (protein quantification); (2) the transformation of dietary protein into body protein content, measured separately in the head, thorax, and abdomen of honey bees (protein quantification and metabolomics); (3) the impact of these diets on the spring dynamics of *DWV* and *SBV* relative expression ratio (transcriptomic analysis); and (4) the effects of the diets on immunity, including the expression of genes related to ROS enzymes (*SOD1*, *SOD2*, and *Trxr1*), innate immunity pathways (Toll: *spz* and *dorsal 1*; Imd: *PGRP-LC* and *relish*; and JAK/STAT: *domeless*), antimicrobial peptide genes (AMPs: *defensin 2* and *apid 1*), and development and nutrition-related genes (*vg*, *JHAMT*, and *TOR1*) in overwintered individual honey bees (transcriptomic analysis). This study ultimately aims to identify standard molecular effects and nutrient markers. Six diet-related marker genes were selected, and they were employed to select the optimal pollen substitute diet for honey bees in comparison to the positive control (Control-P). This research emphasizes the impact of diets on genes associated with the defense system and nutrition-related development, laying the groundwork for establishing molecular markers to assess animal nutrition status. These markers could help in choosing alternative diets and alleviating seasonal malnutrition in animals in agriculture, thus enhancing our quality of life in a changing climate.

## 2. Results

### 2.1. Protein Content in Diets and Its Transformation into Honey Bee Body

The top two best diets were identified based on their high protein content and digestibility in honey bees: Control-P (bee bread) and Diet 2 (Figure 1A). 

Control-P significantly stimulated (*p* < 0.05) protein elevation in the head and thorax more than in the abdomen compared to other overwintered honey bees after a 20 day feeding experiment. Conversely, Diet 2 significantly stimulated (*p* < 0.05) more protein elevation in the abdomen than in the head and thorax (Figure 1B,C). Moreover, similar patterns were observed between proteins in the diet and the digestibility of proteins originating from different diets (Figure 1A). To confirm this, the data were analyzed using a Pearson correlation, revealing a very high correlation (r = 0.974; *p* < 0.05) between protein digestibility in honey bees and the protein concentration in the diet. To quantify this relationship, the protein digestibility was divided by the protein content in each diet. This resulted in values ranging from 3.05 to 3.57 overall for the experimental diets, bee bread, and the commercial diet. This suggests that diet digestibility can be predicted based on the known soluble protein content. However, the index of protein transformation for Diet 2 was 4.9, higher than that of the other diets (Figure 1A), indicating potentially notable properties unique to this diet.

### 2.2. Influence of the Diets on Relative Expression Ratio of the Virus in Honey Bee Colonies in Spring

The dynamics of the sac brood virus (SBV) and deformed wing virus (DWV) loads were examined in honey bees sampled during the spring. The SBV relative expression ratio was not detected, whereas the DWV relative expression ratio was lower in honey bees from colonies fed pollen substitute diets both before and after dieting (Figure 2A) on 16 February and 8 March, compared to honey bees from Control-P colonies, indicating the superior performance of pollen substitute diets over overwintered bee bread. However, with the appearance of natural pollen and the increased activity of honey bee flight on 28 March, only honey bees from Diet 2 colonies maintained a lower DWV load than those from Control-P and other colonies (Figure 2B). 

Next, on 20 April, we examined the long-term effects of diets on short-lived honey bees, which were reared a month prior by long-lived honey bees that had been fed diets within the colony. We observed a similar trend in DWV load among nurse honey bees from Diet 2 colonies, although the virus was detected in foragers as well. This indicates a task-dependent immune response within the same hive environment and suggests that pollen substitute diets may have negligible or no long-term effect.

### 2.3. Selecting of the Nutrition-Related Markers Based on Statistical Scores

The honey bees from different dieting groups were identified based on a comparison of the 18 variables, including the 13 gene expressions via agglomerative hierarchical clustering (AHC) (Figure 3A). The silhouette index (0.164) determined that AHC was feasible. Next, the same dataset was trained by principal component analysis (PCA) (Figure 3B), where the first principal component (F1) explained a significant portion of the dataset’s variability, accounting for 49.34%. 

Variables derived from the PCA method were ranked depending on their decreasing factor loadings, and the top 5 were *SOD2*, *SOD1*, *dorsal 1*, *JHAMT*, and *TOR1*, which appear to be markers. Notably, honey bees from the Control-P and Diet 2 groups clustered closely together in both AHC and PCA analyses, predicting their physiological similarity.

The elastic net regression method was utilized to predict interactions in four models between four protein-related dependent variables (protein concentrations in the diet and in the head, thorax, and abdomen of honey bees) and a gene expression dataset. The four models constructed using elastic net regression were compared based on their mean squared error (MSE), with Model 1 exhibiting the lowest MSE (1.983), indicating its superior predictive accuracy. Models 1 and 2 demonstrated the highest R-squared value (0.935 and 0.915, respectively), suggesting that they explain a larger proportion of the variance in the target variable compared to the other models. However, the analysis ranked gene expression variables similarly across all models, with *TOR1*, *SOD1*, and *defensin 2* genes being the predominant variables. The *vg* gene was associated with proteins in the head, thorax, and abdomen, while the *JHAMT* and *Trxr1* genes were associated with diet and abdomen proteins (Figure 4).

Altogether, six marker gene expressions were visualized in the circular plot to characterize the physiology of honey bees fed with different diets (Figure 5A). We compared honey bees from previously selected experimental Diet 2 colonies using nutrition-related markers with honey bees from Control-P colonies. This comparison was made because the proteins in bee bread (diet in Control-P colonies), as well as their digestion and transformation, were found to be the highest. The honey bees, while being fed Diet 2, exhibited the upregulation of one nutrition-related marker gene, *defensin 2*. However, the expressions of other genes after feeding tended to decrease, approaching those of honey bees from Control-P colonies. The expression of the *JHAMT* gene after feeding showed no significant differences between honey bees from Control-P and Diet 2 colonies, while the expressions of *TOR1*, *SOD1*, and *Trxr1* overlapped (Figure 5B). However, the expression of the *vg* marker gene in honey bees from Diet 2 colonies during dieting was decreased compared to honey bees from Control-P colonies. So, the effectiveness of Diet 2 on honey bee nutrition status was found to be comparable to that of the Control-P diet. However, the hyper-, normal-, and hypo-levels of gene expressions have not been standardized yet, and we lack a methodology to establish a universal scale. 

### 2.4. Influence of the Diets on the Defense System of Overwintered Honey Bees

The honey bees from experimental colonies were compared with honey bees from Control-P (positive control) and Control-N (not normal) colonies. Honey bees from Control-P colonies only received combs with natural honey and bee bread. Honey bees from Control-N colonies were switched from “experimental” to “not normal” colonies for analysis because they exhibited symptoms of chalkbrood from the end of March, and they died at the end of May. However, honey bees from Control-N colonies were not “negative control”, meaning malnutrition, because they received a pollen substitute diet, of which components are listed in Table 1.

The data before and after the experimental dieting represented the initial and the diet-stimulated innate immunity pathways on 16 February and 8 March, respectively. Results are visualized in Figure 6 to illustrate the dynamics of ten gene expressions: *spz* and *dorsal-1* (Toll pathway), *PGRP-LC* and *relish* (Imd pathway), *domeless* (JAK/STAT pathway), *defensin2* and *apid1* (antimicrobial peptides, AMPs), and *SOD1*, *SOD2*, and *Trxr1* (antioxidative defense genes coding ROS enzymes). Comparison was conducted using one-way ANOVA and Duncan post hoc test (*p* < 0.05). 

The dynamics of the standard molecular effect ROS enzymes genes in response to dieting included a decrease in *SOD1* expression and an increase in *SOD2* expression compared to the honey bee condition before and after dieting (Figure 6A,B). The effect in the *Trxr1* gene was unclear (Figure 6C). 

The dynamics of the standard molecular effect on the response of innate immunity genes at the recognition steps in the Toll, Imd, and JAK/STAT pathways related to dieting were the same for all honey bees except those from Control-N colonies. Even though significant differences between honey bees were initially calculated on 16 February, after dieting on 8 March, there were no differences. The molecular effect in the dynamic response of innate immunity pathways (Toll, Imd, and JAK/STAT) to the dieting of honey bees from Control-N colonies showed a significant increase in expression during the recognition step (*spz*, *PGRP-LC*, and *domeless*, respectively) and AMP (*apid 1*), which was not observed in other honey bee groups. This suggests a potential immune disturbance and indicates issues in honey bees from Control-N colonies compared to others, where no significant differences were observed after dieting (Figure 6D–F). 

However, the subsequent dynamic response in all honey bees after dieting showed a general increase in *dorsal 1* and *relish* gene expressions at the recognition step of the Toll and Imd pathways of innate immunity (Figure 6G,H), as well as *defensin 2* marker gene expression for antimicrobial peptides (AMPs) (Figure 6I). However, the dynamics of the *defensin 2* gene were different in honey bees from Diet 1 colonies, where the direction of the trend was opposite (Appendix A).

Importantly, the honey bees from Diet 2 colonies did not belong to the same statistically significant group in marker gene expressions (*SOD1*, *Trxr1*, and *defensin 2*), but they exhibited gene expression patterns closer to those of honey bees from Control-P colonies compared to other honey bees in terms of ROS enzymes and innate immunity gene expressions.

### 2.5. Influence of the Diets on the Nutrition Rate of Overwintered Honey Bees 

The honey bees from experimental colonies were compared with those from Control-P (positive control) and Control-N (not normal) colonies using the same analysis as in Section 2.4. The focus was on the expression of genes such as *vitellogenin* (*vg*), which responds to nutrition, the *JHAMT* gene, positively correlated with juvenile hormone (JH) biosynthesis [28], and the *TOR1* gene, a nutrient-sensing kinase involved in mobilizing nutritional resources from tissues [29]. These genes were previously selected as markers.

The dynamic response of three genes was assessed in gene expression differences after and before dieting (Figure 7A–C and Appendix A). The overall trend in the dynamic response of the *TOR1* gene to dieting was a decrease. However, other genes did not reveal any consistent trends. We visualized the co-repressions of *vg* and *JHAMT* in honey bees in Control-N and Diet 1 colonies, corresponding to the aging of honey bees with a decrease in *vg* and an increase in *JHAMT*. 

Importantly, the honey bees from Diet 2 colonies belong to the same statistically significant group in marker gene expressions *JHAMT* after stimulation, but they exhibited gene expression patterns closer to those of honey bees from Control-P colonies compared to other honey bees in terms of *vg* and *TOR1* gene expressions.

## 3. Discussion

Dietary proteins have no nutritional value until they are hydrolyzed by proteases and peptidases into amino acids, dipeptides, and tripeptides. These components provide essential nitrogen and sulfur for organisms and cannot be replaced by carbohydrates and lipids [30]. The hydrolysis process occurs in the lumen of the small intestine in vertebrates [30] and in the midgut of honey bees [31]. Protein digestion through hydrolysis is crucial because highly digested food requires a smaller amount compared to less-digested food, yet there are still gaps in this area [10]. 

Natural honey bee food, bee bread (Control-P), demonstrated a significantly high protein concentration of 20.59%, with 73.56% of it being digested by honeybees. Experimental data slightly surpassed findings in the literature, which reported an 18.3% protein content in bee bread [32] and a 70% digestion rate of soluble protein in pollen [33], which was similar to the 76% digestibility rate in bee bread [10]. Bee bread showed higher digestibility than pollen, likely due to fermentation in bee bread, which increases digestibility [34]. Using the same analysis, Diet 2 was initially chosen from the studied pollen substitute diets because it contained higher levels of water-soluble proteins (8.81%), of which 43.54% were digested by honey bees, showing a high correlation between them. Unsurprisingly, bee bread, used in this research as a positive control, emerged as the optimal natural diet, as its protein content and digestibility surpassed those of the diets examined in this research.

Furthermore, proteins digested from bee bread and subsequently from Diet 2 were found to be more effective than those from Diet 1 and Megabee diets in terms of accumulating in honey bee body proteins (in the head and abdomen). Head proteins play a crucial role in the development of hypopharyngeal glands in nurse honey bees, which secrete royal jelly to feed the queen and larvae [35]. Abdomen proteins without a digestive system represent the fat body protein content. Previous studies have reported the effects of high-protein nutrition on increasing fat body mass [36], which was identified as an indirect indicator of individual bee immunocompetence [6,20]. So, Diet 2 was the best in the metabolic transformation from diet to head and abdomen proteins in long-lived honey bees compared to Diet 1 and Megabee. 

Because deformed wing virus (DWV) is the most prevalent virus in arthropod species [37,38] and is related to *Varroa* infestation in honey bees, the detection of this virus is the most common method to simply check honey bee health. However, regarding their conclusions, researchers are divided into two groups. One group found that workers from colonies fed a natural pollen diet had significantly lower DWV titers than those fed an artificial diet [39], as well as diets high in fat content [38] and a grape pomace diet [40]. Other researchers did not find clear patterns in DWV or sac brood titers with nutrition [33,41,42], but DeGrandi-Hoffman et al., in 2010 [33], suggested that pollen substitutes can help prevent disease. Continuing this analysis, it was found that the DWV load was lower in long-lived honey bees on 8 March when fed all-pollen substitute diets compared to honey bees from the Control-P colony (bee bread). However, the low levels of DWV in short-lived honey bees on 28 March were observed in honey bees fed only one Diet 2 from studied pollen substitute diets. Therefore, we cannot definitively state the stable effect of pollen substitute diets against the DWV virus, but the effect of Diet 2 appeared to be more efficient than others in early spring. Also, this suggests that fresh pollen, fresh bee bread, and overwintered bee bread may have varying effects on honey bee responses to the DWV virus. Additionally, we cannot determine the long-term effect of early spring feeding on DWV resistance by 20 April. The nurses (one month later without diet stimulation) from all colonies had low DWV loads, but foragers exhibited both high and low DWV loads in different colonies, indicating a task-related effect. Curiously, DWV was low in both nurse and forager honey bees from Control-N colonies, which were infected by chalkbrood and weakened during spring, suggesting that the relative expression ratio of the virus DWV and SBV showed no comprehensive interaction with chalkbrood.

Nevertheless, in Control-N colonies on 8 March, we observed a significant upregulation in the recognition steps of the Toll, Imd, and JAK/STAT pathways, as represented by *spz*, *PGRP-LC*, and *domeless*, as well as in *apid 1* gene expressions (AMP). Therefore, innate immunity genes predicted the weakened health of honey bees on 8 March, anticipating the observed chalkbrood symptoms on 28 March. Thus, genes from the recognition step of the innate immune response can be recommended for future research to identify the health problems in honey bees related to chalkbrood and similar diseases.

Gene expression also indicates how an organism responds to nutrition at the molecular level. The nutritional marker genes that responded to dieting included *defensin 2*, *SOD1*, *Trxr1*, *JHAMT*, *TOR1*, and *vg*. Some of these genes, previously reported to be related to nutrition, are involved in vitellogenin synthesis [43], antioxidant enzymes [44], and immune function [45]. Although studied in short-lived honey bees, our research was conducted on long-lived honey bees and yielded similar results. Additionally, *vg*, *JHAMT*, and *TOR1* genes responded not only to nutrition [46] but also to the transformation of insects from obligatory winter diapause to spring reproductive activity [47,48]. Based on this, a molecular model was developed to elucidate the transition of long-lived honey bees from diapause to development (Figure 8).

Because worker honey bees do not lay eggs under normal conditions, as expected, their gene expression model differed from that of *Galeruca daurica* [48] in terms of high, but not low, *vg* gene expression [49] and additionally showed a positive correlation with *TOR1* expression, confirming the findings of [46]. However, *JHAMT* gene expression remained low, as typically observed in diapausing insects [50]. Also, the nutritional stimulation of worker honey bees at the end of diapause was associated with an increase in *JHAMT* expression, but a decrease in *TOR1* and *vg* genes (Figure 8).

Feeding on an artificial diet resulted in the upregulation of the *SOD2* gene (ROS enzyme) in the general innate immunity response. Additionally, *spz* (Toll pathway, recognition step) and *TOR1* (TOR pathway) were downregulated in all honey bees. This phenomenon mirrors the findings of a diet experiment conducted on fish. In that study, increasing the dietary tryptophan (Trp) concentration from 4.8 to 6.8 g/kg resulted in enhanced activities of total antioxidant capacity (T-AOC) and superoxide dismutase (*SOD*, ROS enzymes). Conversely, mRNA expression levels of the target of rapamycin (TOR pathway) and Toll-like receptors in the intestine (Toll pathway) were significantly downregulated [51]. Therefore, a balanced diet with a moderate amount of Trp (4.8 g/kg) was observed to be effective for fish. Considering this, it is essential to ensure that the species-specific protein content in the diet is optimal for animals, neither too high nor too low. This aspect warrants further study in the future.

In our study, we employed nutritional markers. However, the hyper-, normal-, and hypo-levels of gene expression have not yet been standardized, and we lack a methodology to establish a universal scale. Therefore, we cannot conclusively determine whether the stimulation of genes to upregulate or downregulate had a negative or positive effect on honey bee health. This will be a challenge for future research.

## 4. Material and Methods

### 4.1. Experimental Honey Bees

Honey bees (*A. m. ligustica*) were sourced from fifteen colonies at the Gangneung apiary in the Republic of Korea in 2023. Managed by a professional beekeeper, experimental colonies adhered to standard beekeeping practices to prevent nutrient stress. Samples for analysis were collected from colonies showing no clinical symptoms of diseases or infestation by *Varroa destructor*. The study involved three biological replications for each diet. The impact of the diet was examined in honey bees sampled on 16 February (owb, overwintered honey bees) and 8 March (owb+diet, overwintered honey bees under dietary conditions), which were 20 days before and after commencing the diet experiment, respectively (Figure 9). 

During this period, spring generations of honey bees were raised but had not yet emerged. After emerging, they were sampled for ongoing disease detection. The newly emerged honey bees were collected on 28 March, the final day of the diet experiment when diets were discontinued. After 20 days, these same honey bees were expected to have transformed into foragers and were then collected at the hive entrance with pollen loads. Nurses, responsible for caring for larvae, were also sampled on 20 April.

The data on honey bees from Diet 1, Diet 2, and Megabee colonies were compared to those of honey bees from Control-P colonies, which received natural honey bee food without restrictions and also achieved the highest management scores during and after the experiment in 2023. Additionally, the further validation of the conclusions was conducted by comparing them with honey bees from Control-N colonies (“not normal”), which also received a diet at the same time as the other colonies; however, symptoms of chalkbrood were detected from 29 March. Symptoms increased before the eventual collapse of the colonies at the end of May. A negative control, where honey bees experienced starvation, was not used to avoid the early collapse of colonies, which could have prematurely ended the field experiment. 

### 4.2. Preparation of Diets

The pollen substitution diets included protein, carbohydrates, minerals, and fats, and their nutrition was described previously [18,19]. These items were reasonably priced in the local market, and special components were Soytide (CJ Global Food and Bio Company, Seoul, Republic of Korea), apple juice (Jaan Company, Dubai, United Arab Emirates), and chlorella powder (Cheonil Herbal Medicine, Seoul, Republic of Korea). The Supplemental Diets listed in Table 1 were created. 

Each diet was tested with three replications in the field for three pollen substitute diets, including naturally presented bee bread in colonies as a positive control (Control-P). The various Supplemental Diets were prepared separately first by measuring known quantities and carefully blended by hand. The commercial diet Megabee (Castle Dome Solutions, Helena, AR, USA), a blend of plant-based proteins without pollen that is widely used by beekeepers in the USA, was also tested.

### 4.3. Protein Concentration

The protein concentrations of the diet, head, thorax, hindgut, and abdomen were measured separately. All steps of sample preparation were conducted using iced chemicals and tissues. Instruments were treated with 70% ethanol between the dissections of samples from different colonies. Three honey bees from each colony were dissected, separating the heads, thoraxes (without wings and legs), and abdomens (without the digestive system, with hindguts removed for disability analysis). Samples from the three bees per colony were pooled in a 1.5 mL tube using the Ultra Grinder B kit (Taeshin Bioscience Co., Ltd., Busan, Republic of Korea). Samples were prepared by weighing the corresponding parts of the honey bees or 200 mg of diet on a balance. Next, an amount of 0.25 M Tris HCl buffer at pH 7.5 was measured to obtain 20% of the tissue in sample. The honey bee tissues or diet were then ground in this tube using a sterile disposable homogenizer included in the Ultra Grinder B kit, along with 100 µL of buffer. The mixture was later diluted to a 20% solution using the same buffer and centrifuged at 13,000× *g* for 30 s, and the supernatant was used to measure the protein concentration.

The water-soluble total protein concentration was determined using the colorimetric method with the Pierce™ BCA Protein Assay Kit (bicinchoninic acid, BCA) (Thermo Fisher Scientific Inc., Waltham, MA, USA) following the manufacturer’s instructions. The eight-point calibration curve was performed in duplicate from the bovine serum albumin (BSA) included in the kit in the concentration range from 0 to 2000 µg/mL (Appendix A). 

The analyses were performed in triplicate. Samples in amounts of 25 µL of diet, head, and thorax and 1 µL of gut and abdomen samples were mixed with the working reagent on a sterile polystyrene black plate with 96 wells (SPL Life Science Co., Ltd., Pocheon, Republic of Korea) and kept under 37 °C for 30 min in the incubator. After that, the absorbance at 562 nm was measured on the multimode reader BioTek Synergy HTX (Agilent, Santa Clara, CA, USA) using Agilent BioTek Gen 5 Software (Agilent, CA, USA). The standard curve was used to determine the protein concentration of each unknown sample [52] and recalibrated for one unit of the honey bee’s head, thorax, and abdomen.

### 4.4. Protein Digestion

The approximate protein digestibility was calculated using Equation (1).
Approximate protein digestibility (%) = (A − B)/A × 100(1)
where A is the total water-soluble proteins in the diet, and B is the total water-soluble proteins in the hindgut [53,54].

### 4.5. RNA Extraction and cDNA Synthesis

A total of 300 honey bees (20 winter bees from each colony) *A. m. ligustica* from each sampling date were collected in labeled tubes and stored at −80 °C. Three honey bees from each colony were analyzed. Total tissue RNA was extracted from the whole bodies of nine randomly selected bees in each dietary group using a Qiagen RNeasy Mini Kit (#74104; Qiagen, Valencia, CA, USA). The RNA concentration and purity were quantified using OD260/OD280 values between 1.8 and 2.0. Next, reverse transcription was performed using an RNA to cDNA EcoDryTM Premix (Oligo dT) kit (Takara, Osaka, Japan). The reverse transcription reaction mixture included 50 ng/µL total RNA (with the clear volume calculated for each sample) and RNase-free water for a total volume of 20 µL. Reverse transcription was conducted at 42 °C for 60 min, followed by heating at 70 °C for 10 min.

### 4.6. Quantitative Real-Time PCR 

Relative expression was measured for ten genes of defense system (*spz*, *dorsal-1*, *PGRP-LC*, *relish*, *domeless*, *defensin-2*, *apid-1*, *SOD*, *SOD2*, and *Trxr1*), two virus-related genes (*DWV* and *SBV*), and three genes related to nutrition and development (*vg*, *TOR1*, and *Juvenile Hormone Acid Methyltransferase*, *JHAMT)*. The housekeeping gene *β-actin* was used as an endogenous control. The PCR primer sequences are shown in Appendix A. The reaction conditions were optimized. Quantitative real-time PCR (qRT-PCR) was conducted using an AccuPower 2X GreenStarTM qPCR Master Mix (BIONEER, Oakland, CA, USA) on an AriaMx Real-Time PCR System (Agilent Technologies LDA, Penang, Malaysia). The qRT-PCR reaction volume of 20 µL included 2 µL of template cDNA, 10 µL of 2X GreenStar Master Mix, 1 µL of upstream and downstream primers (5 pM/µL), and 6 µL of ddH_2_O. Each sample was technically replicated three times.

The qRT-PCR amplification conditions were as follows: an initial denaturation at 95 °C for 10 min, followed by 40 cycles of denaturation at 95 °C for 30 s, annealing at 60 °C for 25 s, and extension at 72 °C for 15 s. Data analysis was performed using free analysis software Agilent AriaMx version 2.0. Relative gene expression data were analyzed using the 2^(−ΔΔC(T))^ method [55,56]. 

The confirmation of *β-actin* and amplicons in the RT-PCR products was performed by separating them through electrophoresis in a 1% agarose gel at 100 V for 20 min and analyzing them using a gel documentation system, the Gerix 1010 transilluminator (Biostep GmbH, Reiskirchen, Germany). A 100 bp DNA Ladder (BIONEER, BIO-RAD, Seoul, Republic of Korea) was used as a reference. The confirmation of amplicons in the RT–PCR products was performed by separating them through electrophoresis in a 2% agarose gel at 80 V for 40 min and analyzing them using the same gel documentation system, the Gerix 1010 transilluminator. A Dyne 50 bp DNA Ladder (Cat. No. A701, DYNEBIO, Seongnam, Republic of Korea) was used as a reference.

### 4.7. Statistical Analysis

The statistical analysis, designed as illustrated in Figure 10, was conducted using Microsoft Excel and XLSTAT Life Science version 2023.2.0 (Addinsoft, Paris, France). Specifically, the categorization of honey bees into groups was plotted using agglomerative hierarchical clustering (AHC). AHC was employed to analyze the multivariate dataset containing 18 characters, grouping similar data points into clusters based on their pairwise distances using a bottom-up approach [57]. Additionally, the principal component analysis (PCA) plot was used within the same dataset to capture the majority of the variance in a reduced-dimensional space to identify the best diet near Control-P. Moreover, the interpretation of the PCA plot highlighted the principal components contributing most significantly to the observed patterns, predicting the key molecular markers driving the variation in the data. 

The same dataset was trained in elastic net regression analysis to confirm the predicted markers in PCA and informed decision-making against the dependent variable by score. The elastic net regression model demonstrated robust performance in handling multicollinearity and selecting relevant predictors, leading to improved predictive accuracy compared to traditional regression techniques [58]. The objective was to identify the combination that minimizes the model’s error metrics, such as mean squared error (MSE) or R squared. MSE measures the average squared difference between the actual values and the values predicted by the model. Among several models, the one with the lower MSE is generally preferred. R-squared measures the proportion of the variance in the dependent variable (target) that is explained by the independent variables (features) in the model. R-squared ranges from 0 to 1, where 0 indicates that the model does not explain any of the variability in the target variable, and 1 indicates that the model explains all of the variability.

Pairwise correlation analysis was conducted using the Pearson method [59] to identify significant relationships between the traits (*p* < 0.05). The correlations were interpreted based on the guidelines provided by Hinkle et al. in 2003 [60], categorizing correlations as very high positive (negative) correlation (±0.90 to 1.00), high positive (negative) correlation (±0.70 to 0.90), moderately positive (negative) correlation (±0.50 to 0.70), low positive (negative) correlation (±0.30 to 0.50), and negligible correlation (±0.00 to 0.30) [59]. The mean, standard deviation values, and normal distribution were calculated using the descriptive statistics module. The visualization of gene expression was performed using the heatmap module. For multiple comparisons of variables between groups of honey bees, ANOVA was employed to test overall effects, followed by the Duncan post hoc test (*p* < 0.05).

## 5. Conclusions

The development and evaluation of new artificial diets involve testing the sensitivity of the honey bee model to protein deficiency through nutritional markers. Over the past decades, cumulative studies have enhanced our understanding of alternative diets for humans, animals, and honey bees, as well as their impact on health. High protein content, digestibility, and nutrient transformation in the body were key factors in selecting the best artificial diet (Diet 2), which closely resembled the characteristics of honey bees that were fed natural food (bee bread). Using machine learning, six molecular markers were selected to investigate the diet’s impact on health and development and then tested by comparing experimental honey bees with those that were fed bee bread. Honey bees fed Diet 2 exhibited similarities to those fed bee bread. However, lacking a malnutrition control, we refrained from clearly identifying the up- and downregulation of marker gene expression. Additionally, the health of honey bees showing symptoms of chalkbrood was uniquely stimulated compared to others, indicating differences in dietary stimulation between health-deficient and healthy honey bees. Furthermore, we proposed a molecular model to elucidate the transition of long-lived honey bees from diapause to development, triggered by nutrition. This research underscores the influence of diets on genes associated with the defense system and nutrition-related development, laying the foundation for establishing molecular markers to assess animal nutrition status. These markers could aid in selecting alternative artificial diets and mitigating seasonal malnutrition in agricultural animals, thereby improving our quality of life in a changing climate.

## Figures and Tables

**Figure 1 ijms-25-04271-f001:**
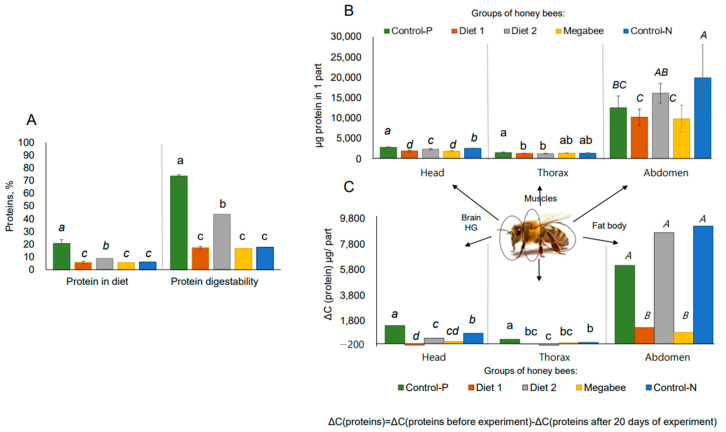
Protein content in diets and its transformation into honey bee body in overwintered honey bees after dieting in early spring (sampling date: 8 March 2023). One-way ANOVA, Duncan post hoc test (*p* < 0.05). (**A**)—Percentage of proteins in diets and their digestibility. (**B**)—Protein concentration in different parts of the honey bee body; gut was removed from the abdomen. (**C**)—Change in protein concentration during experimental feeding (transformation of protein from diet to honey bee body); gut was removed from the abdomen. In (**A**), italicized lowercase letters denote significant differences in protein content among diets, while lowercase letters represent significant differences in protein digestibility among honey bees. In (**B**,**C**), italicized lowercase letters denote significant differences between samples in the head, lowercase letters represent significance between samples in the thorax, and italicized uppercase letters indicate significance between samples in the abdomen surrounding the current chart.

**Figure 2 ijms-25-04271-f002:**
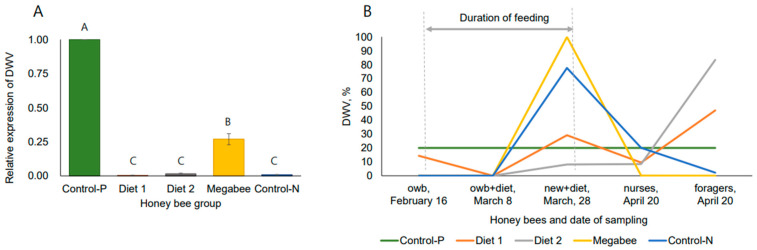
Deformed wing virus (DWV) load in honey bee colonies in spring. (**A**)—Relative expression level of DWV in overwintered honey bees after dieting in early spring (sampling date: 8 March 2023). One-way ANOVA, Duncan post hoc test (*p* < 0.05). (**B**)—Dynamics of DWV relative expression ratio before (sampling date: February 16), during (sampling dates: 8 and 29 March), and after (sampling date: 20 April) the experiments to investigate the long-term effect of the diets. owb—the long-lived overwintered bees; owb+diet—long-lived overwintered honey bees after dieting; new+diet—short-lived new honey bees emerged to replace overwintered honey bees; nurses—short-lived honey bees provided nursing duties; foragers—short-lived honey bees returning to the hive with pollen loads. In (**A**), uppercase letters indicate significant differences in DWV expression among honey bees from different dieting groups.

**Figure 3 ijms-25-04271-f003:**
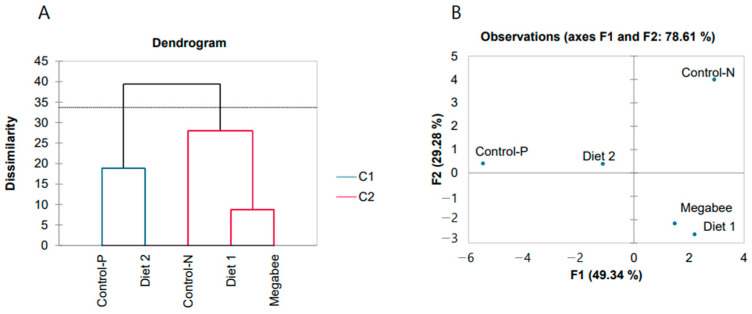
Discrimination of overwintered honey bees was conducted based on 18 variables, which included protein content in the diet, protein digestion, and gene expressions related to the defense system and nutrition. (**A**)—Agglomerative hierarchical clustering. (**B**)—Principal component analysis (sampling date: 8 March 2023). In (**A**), C1 and C2 represent two different clusters identified by the AHC analysis. In (**B**), F1 and F2 represent the first and second principal components, respectively, which capture the maximum variability in the dataset along orthogonal axes.

**Figure 4 ijms-25-04271-f004:**
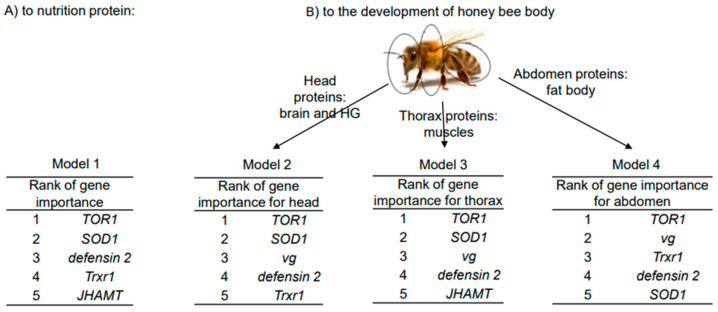
Top 5 molecular marker genes selected by elastic net regression method for each dependent variable of overwintered honey bees in four models (sampling date: 8 March 2023). (**A**)—Model with the dependent variable as nutrition proteins. (**B**)—Models with the dependent variables as proteins of honey bee body parts (head, thorax, and abdomen).

**Figure 5 ijms-25-04271-f005:**
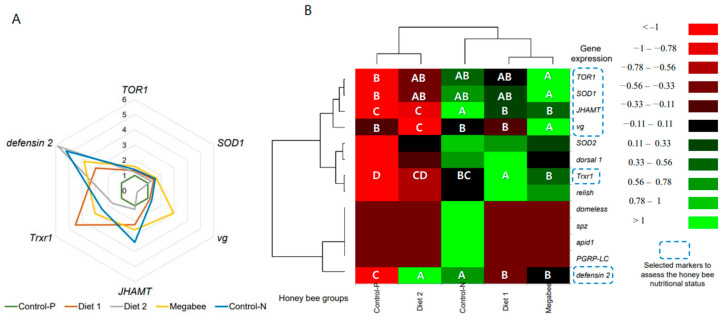
Gene expression of selected top molecular markers in response to nutrition and head/thorax/abdomen proteins of overwintered honey bees (sampling date: 8 March 2023). (**A**)—Circular plot displaying the expressions of six marker genes. (**B**)—Heat map with clustering by honey bee groups (horizontal) and 13 gene expressions (vertical), highlighting the marker genes with blue dotted squares. Boxes marked with (A) represent statistically significant upregulation, and those marked (B–D) represent downregulation due to nutrition response, with a *p*-value less than 0.05. One-way ANOVA and Duncan post hoc test were used.

**Figure 6 ijms-25-04271-f006:**
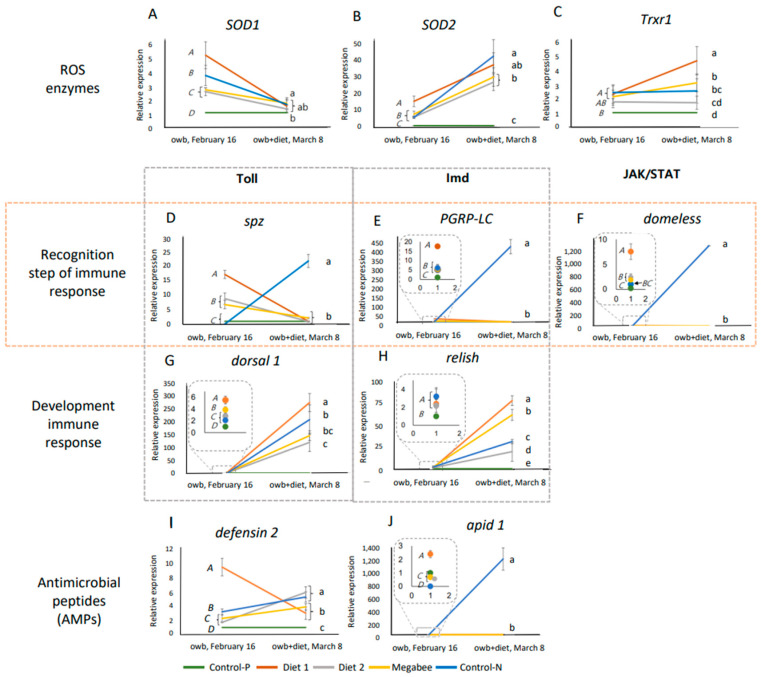
The dynamics of defense system gene expressions in overwintered honey bees are represented by means and standard deviations before (owb, 16 February) and after the diet experiment (owb+diet, 8 March). (**A**–**C**) represent the gene expressions related to ROS enzymes. (**D**,**G**) represent the gene expressions related to Toll pathway. (**E**,**H**) represent the gene expressions related to Imd pathway. (**F**) represents the gene expression related to JAK/STAT pathway. (**I**,**J**) represent the gene expressions related to antimicrobial peptides (AMPs). One-way ANOVA and Duncan post hoc test (*p* < 0.05) were conducted. Owb represents overwintering bees, while owb+diet represents overwintered honey bees after dieting. In each chart of Figure 6, italicized uppercase letters indicate significant differences between overwintered honey bee samples collected on February 16 before dieting. Lowercase letters on the same chart denote significant differences between honey bee samples after the dieting on 8 March. The dotted line squares in (**E**–**H**,**J**) indicate magnified areas on the chart, highlighting significant differences in gene expression among honey bees sampled on the same day.

**Figure 7 ijms-25-04271-f007:**
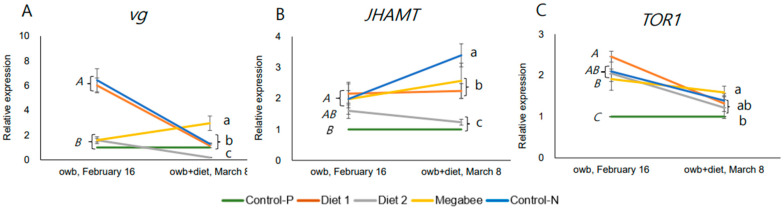
Dynamics of nutrition marker gene expressions in overwintered honey bees represented by means and standard deviations before (owb, 16 February) and after the diet experiment (owb+diet, 8 March). (**A**–**C**) represent the gene expressions related to nutrition and development. One-way ANOVA and Duncan post hoc test (*p* < 0.05) were conducted. owb—overwintering bees; owb+diet—overwintered honey bees after dieting. In each chart of (**A**–**C**), italicized uppercase letters indicate significant differences between overwintered honey bee samples collected on 16 February before dieting. Lowercase letters on the same chart denote significant differences between honey bee samples after the dieting on 8 March.

**Figure 8 ijms-25-04271-f008:**
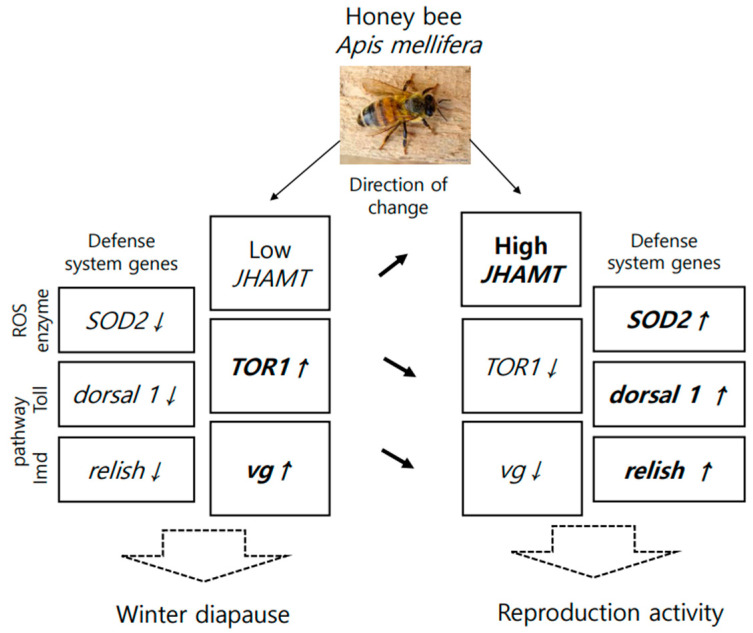
A molecular model of the transition of long-lived honey bees from diapause to development, triggered by nutrition. Arrows presented the direction of changes gene expression, this noticed above arrows.

**Figure 9 ijms-25-04271-f009:**
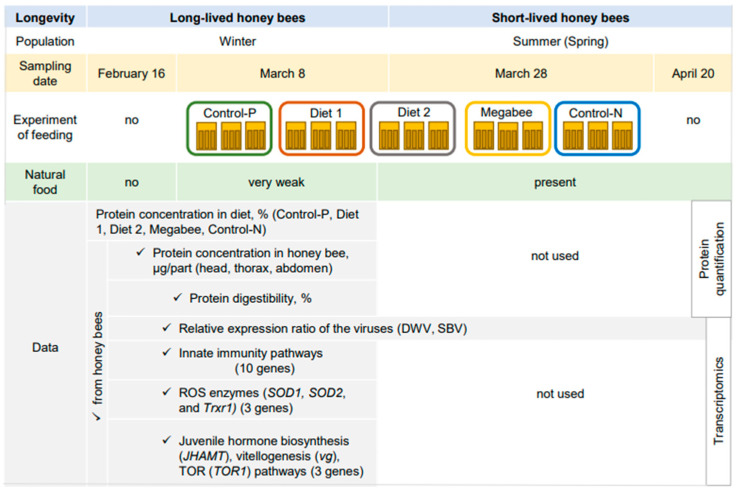
Design of experiment.

**Figure 10 ijms-25-04271-f010:**
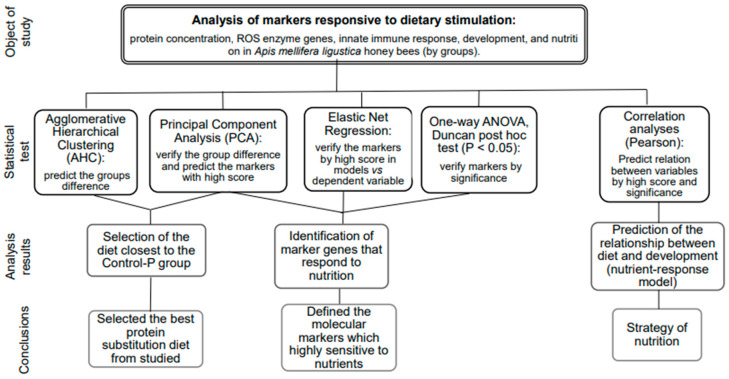
The statistical tests and criteria used for selection of markers and making decisions.

**Table 1 ijms-25-04271-t001:** Composition of pollen substitute diets with constant ingredients (%).

Ingredients	Diet 1 (Soytide)	Diet 2 (SAC)	Control-N (Apple Juice)
Brewer’s yeast	39.69	39.69	39.69
Egg yolk	2.21	2.21	2.21
Defatted soybean powder	-	-	2.21
Sugar	35.36	35.36	35.36
Boiled water	5.16	7.16	7.16
Canola oil	1.01	1.01	1.01
Cellulose	0.88	0.88	0.88
Wheat bran powder	0.88	0.88	0.88
Multiple vitamins	0.44	0.44	0.44
L-methionine	0.1	0.1	0.1
L-lysine	0.24	0.24	0.24
Citric acid	1.85	1.85	1.85
IMP	0.0002	0.0002	0.0002
GMP	0.0002	0.0002	0.0002
Tangerine juice	10	4	4
Soytide powder	2.21	2.21	-
Apple juice	-	4	4
Chlorella powder	-	0.08	-

## Data Availability

The data presented in this study are available on request from the corresponding author.

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
