# Peer review of "The Effects of Artificial Diets on the Expression of Molecular Marker Genes Related to Honey Bee Health"

_ijms, 2024, doi:10.3390/ijms25084271_

Round 1
Reviewer 1 Report
Comments and Suggestions for Authors
General remarks
The entire MS is well written with enough information for anyone who wants to repeat the experiment or any subset of it. Importantly, if a commercial kit is used then the authors give all necessary details for anyone who wants to obtain it from the local state market.
Specific remarks
1. Line 222: The authors state that ‘To prove the clustering, the Principal Component Analysis (PCA) plot was used’. PCA is not a verification method for cluster analysis. The two methods reveal different aspects of the data. PCA ordinates the samples in a uni – di – tri – multidimensional significant space while cluster analysis finds the similarities among samples. Please rephrase this point of the text.
2. Lines 229-241: Reference of Wang et al., 2022 states that the ‘Elastic Net Regression’ model outperforms all other penalized regression models if the number of variables are [sic] far more than the observations -i.e. ‘elastic net is particularly useful when the parameters are far more than the observations’. Is this the case of the experiment described in the MS. Please rephrase.
3. Figure 5B: A better writing style of the explanation of x-coordinate labels would be very helpful, e.g. new+diet := short lived new honeybees emerged to replace overwintered honeybees. Please keep lowercase letters in the explanation panel.
4. Figure 6Α: What are the C1 and C2 on the left side of the dendrogram?
5. Figure 6B: If the authors insist on presenting an ordinogram of PCA then they should write in the text the meaning attached to the two Principal Axes.
6. I am unable to find any text passage explaining the way the authors measured the viral load.
My recommendation for this MS is ‘Minor Revision’ after addressing all the points above.
Author Response
Dear Editor-in-Chief, and Reviewer 1!
Thank you for the time and effort spent reviewing our manuscript number ijms-2945124 and suggesting some important points to consider.
Our replies are marked by green color after the symbol R (reply) and question numbers 1 (R1) – 15 (R15) for the Academic Editor, 16 (R16) – 20 (R20) for the first reviewer’s comments, and (22 (R21) – 30 (R29)) for the second reviewer’s comments. Also, we copied these modifications in response for convenience when it was available.
Please, find below the reviewer comments (black) and our response (green) in the attached document named Responce.docx.
We would like to thank the reviewers for their careful and thorough reading of this manuscript and for the thoughtful comments and constructive suggestions, which helped to improve the quality of this manuscript.
We look forward to hearing from you.
Sincerely,
Professor Hyung Wook Kwon
Department of Life Science,
Convergence Research Center for Insect Vectors
Incheon National University
119 Academy-ro, Yeonsu-gu, Incheon, 22012, Republic of Korea
+82-32-835-8090 (work);
+82-10-3379-6727 (mobile);
+82-32-835-0764 (fax).
Email: [email protected]

Reviewer 2 Report
Comments and Suggestions for Authors
The topic of the article is very interesting. The authors used honey bees fed different diets to test some parameters investigated at the transcriptome level could be used as markers of "good nutrition" in the bee.
In the abstract the authors talk about proteomics but in the paper I did not find a proteomic investigation. I ask the authors to remove this disregarded announcement, please.
The introduction completely misses the investigation that has been carried out in recent years by several authors on the effects of diets on enzymes involved in the immune and antioxidant systems and immune system cells. I ask the authors to add this part, please.
Minor comments:
-page 2 lines 89-91 this part belongs to the results section, please move it.
-Figure 2 the title of the x-axis is “concentration” instead of “concentrition”. please change it.
- page 7 lines 248-251 statistical analysis: before applying the anova test was it verified that the data was normally distributed? otherwise a nonparametric test should be used.
-page 8 lines 271-276 this part belongs to the discussion section, please move it and add a bibliographical reference at line 273.
-page 9 lines 298- 308 this part belongs to the discussion section, please move it and add a bibliographical reference at lines 298 and 303.
-pages 13-14 lines 437-445 this part belongs to the aim of the work section, please move it.
-page 14 lines 474-481 This part is also missing from the studies on the effects of diets on enzymes involved in the immune and antioxidant systems and immune system cells. Toll, IMD, Jak/STAT are linked to the phenoloxidase enzyme cascade that results in melanin production, a compound involved in encapsulation and nodule formation. Please elaborate on this topic.
Comments on the Quality of English LanguageMinor editing of English language required
Author Response
Dear Editor-in-Chief, and Reviewer 2!
Thank you for the time and effort spent reviewing our manuscript number ijms-2945124 and suggesting some important points to consider.
Our replies are marked by green color after the symbol R (reply) and question numbers 1 (R1) – 15 (R15) for the Academic Editor, 16 (R16) – 20 (R20) for the first reviewer’s comments and (22 (R21) – 30 (R29)) for the second reviewer’s comments. Also, we copied these modifications in response for convenience when it was available.
Please, find below the reviewer comments (black) and our response (green) in the attached document named Responce.docx
We would like to thank the reviewers for their careful and thorough reading of this manuscript and for the thoughtful comments and constructive suggestions, which helped to improve the quality of this manuscript.
We look forward to hearing from you.
Sincerely,
Professor Hyung Wook Kwon
Department of Life Science,
Convergence Research Center for Insect Vectors
Incheon National University
119 Academy-ro, Yeonsu-gu, Incheon, 22012, Republic of Korea
+82-32-835-8090 (work);
+82-10-3379-6727 (mobile);
+82-32-835-0764 (fax).
Email: [email protected]
